# Spray Dried Levodopa-Doped Powder Potentially for Intranasal Delivery

**DOI:** 10.3390/pharmaceutics14071384

**Published:** 2022-06-30

**Authors:** Xuan Liu, Shen Yan, Mengyuan Li, Shengyu Zhang, Gang Guo, Quanyi Yin, Zhenbo Tong, Xiao Dong Chen, Winston Duo Wu

**Affiliations:** 1Engineering Research Centre of Advanced Powder Technology (ERCAPT), School of Chemical and Environmental Engineering, College of Chemistry, Chemical Engineering and Materials Science, Soochow University, Suzhou 215123, China; xuanliu520@126.com (X.L.); 20217909003@stu.suda.edu.cn (S.Y.); 20204209307@stu.suda.edu.cn (M.L.); 20194009046@stu.suda.edu.cn (S.Z.); xdchen@mail.suda.edu.cn (X.D.C.); 2School of Energy and Environment, Southeast University, Nanjing 210096, China; guogang55@163.com; 3Key Laboratory of Energy Thermal Conversion and Control of Ministry of Education, School of Energy and Environment, Southeast University, Nanjing 210096, China; z.tong@seu.edu.cn; 4Centre for Simulation and Modelling of Particulate Systems, Southeast University-Monash University Joint Research Institute, Suzhou 215123, China

**Keywords:** levodopa-doped nasal powder, spray drying, controlled drug release, strong mucoadhesive force, high deposition efficiency

## Abstract

This work was aimed to develop levodopa (L-dopa) nasal powder to achieve controllable drug release and high nasal deposition efficiency. A series of uniform microparticles, composed of amorphous L-dopa and excipients of hydroxypropyl methyl cellulose (HPMC), polyvinylpyrrolidone (PVP), or hydroxypropyl-β-cyclodextrin (CD), were fabricated by a self-designed micro-fluidic spray dryer. The effects of excipient type and drug/excipient mass ratio on the particle size, morphology, density, and crystal property, as well as the in vitro performance of drug release, mucoadhesion, and nasal deposition, were investigated. Increased amounts of added excipient, regardless of its type, could accelerate the L-dopa release to different extent. The addition of CD showed the most obvious effect, i.e., ~83% of L-dopa released in 60 min for SD-L1CD2, compared to 37% for raw L-dopa. HPMC could more apparently improve the particle mucoadhesion than PVP and CD, with respective adhesive forces of ~269, 111, and 26 nN for SD-L1H2, -L1P2, and -L1CD2. Nevertheless, the deposition fractions in the olfactory region for such samples were almost the same (~14%), probably ascribable to their quite similar particle aerodynamic diameter (~30 μm). This work demonstrates a feasible methodology for the development of nasal powder.

## 1. Introduction

Nasal administration offers a non-invasive pathway to delivery drugs for treating both local and systemic diseases, with several advantages of bypassing the blood-brain barrier, avoiding hepatic first-pass elimination, and reducing side effects [1,2,3]. Particularly aiming at central nervous system (CNS) diseases, including schizophrenia, migraines, Parkinson’s, etc., intranasal drug delivery via the olfactory and trigeminal routes directly to the brain is capable of providing much more effective drug concentration at lower doses and reduced risk for adverse reactions [4,5,6,7]. For instance, levodopa (L-dopa), as the most effective antiparkinsonian agent, is usually administrated orally, but its bioavailability is extremely low (~1%), such that over nearly half of Parkinson’s disease patients experience the OFF episode, during which the symptoms of tremor, stiffness, or slowness of movement dramatically increase [8,9,10]. Nasal delivery of L-dopa has the potential to solve such problems by enhanced local drug concentration in the brain [11,12,13].

At present, most of the commercially available nasal drugs are in liquid form. However, the liquid form is not appropriate for L-dopa, as its dissolved molecule is unstable and tends to be oxidized [14,15]. It was reported that, after storage for 9 days at 30 °C, the L-dopa concentration in the solution of maleic acid (pH = 2.7) decreased to 81.2 ± 0.2%, which highly jeopardized the treatment effect [16]. By contrast, the powder form of L-dopa can be of superior stability, both physically and chemically, and more favorable for achieving controllable drug release to maximize the therapeutic efficacy [13,17]. However, difficulties still arise from the complicated physiological structure of the nasal cavity, including the complex geometry hindering the delivery of the drug to the targeted area, cilia clearance mechanism shortening the residence time of powders, and tightly connected nasal mucosa limiting the drug permeability [2,18], which have to be fully considered for nasal powder development. To tackle such challenges, different powders have been developed by co-micronization of L-dopa with functional excipients via various particle engineering technologies. For instance, Bartos et al. [19] manufactured L-dopa powders co-milled with chitosan or sodium hyaluronate as the mucosal adhesion enhancer, which showed the prolonged residence time in the nasal cavity. Another type of L-dopa powder was fabricated by freeze drying co-dissolved L-dopa with maleic acid and hydroxypropyl-β-cyclodextrin (CD) as the excipients, which was proven to have good physical stability and improved distribution and absorption of L-dopa in the nasal cavity [16]. In addition, other techniques, such as spray drying [13], supercritical fluid techniques [20,21,22], and ionic gel methods [11,23], have been utilized to prepare L-dopa powders. Nevertheless, the L-dopa nasal powders reported in previous relevant studies generally had wide size distribution and/or were severely agglomerated, resulting in large variations in the powder property and performance, even in the same sample batch, which largely impeded the development of high-quality nasal powder formulations. The fabrication of powders with uniform particle size and morphology is necessitated, in order to precisely investigate the structure–performance relationship, i.e., correlating the particle structure attributes with the nasal deposition efficiency and drug release behaviors.

This study was aimed at fabricating L-dopa-doped powders, potentially for nasal administration with controllable drug release, high nasal delivery, and deposition efficiency. A self-designed micro-fluidic spray dryer (MFSD) capable of manufacturing uniform microparticles with tailorable characteristics and functionalities was utilized [24,25,26]. Hydroxypropyl methyl cellulose (HPMC), polyvinylpyrrolidone (PVP), and CD were selected as excipients, which were used as the crystal inhibitor to fabricate the amorphous solid dispersions. For nasal administration, HPMC and PVP were also used as the mucoadhesive agent, while CD was used as the absorption enhancer [15]. The effects of excipient type and drug/excipient mass ratio on particle physico-chemical properties, including the size, morphology, crystalline, and thermal properties, were characterized using scanning electron microscopy (SEM), X-ray diffraction (XRD), Fourier transform infrared spectrometry (FT-IR), and differential scanning calorimetry (DSC). In vitro particle adhesion and L-dopa release behaviors were analyzed using atomic force microscopy (AFM) and a Franz diffusion cell apparatus with simulated nasal fluid, respectively. The nasal delivery and deposition performance of the obtained L-dopa-doped powders were evaluated using a 3D-printed quasi-real human nasal cast.

## 2. Materials and Methods

### 2.1. Materials

Levodopa (L-dopa, 3,4-dihydroxy-L-phenylalanine), hydroxypropyl-β-cyclodextrin (CD, purity > 99%, Mw = 1541.54 g/mol), and hydroxypropyl methyl cellulose (HPMC, purity ≥ USP 2910, 2% viscosity: 3 mPa·s, methoxy: 28–30%, hydroxypropyl: 7.0–12%) were purchased from Yuanye Bio-Technology Co., Ltd. (Shanghai, China) and Aladdin Biochemical Technology Co., Ltd. (Shanghai, China), respectively. Polyvinylpyrrolidone K30 (PVP) was purchased from Solarbio Science and Technology Co., Ltd. (Beijing, China). Porcine gastric mucin was supplied by Kechuang Biotechnology Co., Ltd. (Suzhou, China). Simulated nasal fluid (SNF) was prepared as an aqueous solution containing NaCl (150.0 mM; Macklin, Shanghai, China), KCl (40.0 mM; Macklin, Shanghai, China), and CaCl_2_∙2H_2_O (5.3 mM; Macklin, Shanghai, China) [27,28]. Deionized water (~18.2 MΩ/cm) from a Milli-Q apparatus (Merck, Darmstadt, Germany) was used throughout the experiments.

### 2.2. Preparation of Feed Solution

Various feed solutions were prepared according to the formulation listed in Table 1. In a typical procedure, 0.3 g of HPMC and 0.3 g of L-dopa were dissolved in 99.4 g of deionized water, under stirring (600 rpm) at 40 °C, to obtain a clear solution without any visible substance being present. The obtained feed solution with a mass ratio of L-dopa/excipient = 1:1, namely L1H1, was used for spray drying. The same procedure was employed for other feed solutions, in which the total solid content was kept at 0.6 wt%, while the mass ratios of L-dopa for each excipient were 2:1, 1:1, and 1:2, respectively. In addition, a pure L-dopa feed solution with a mass fraction of 0.3 wt% was also prepared for spray drying.

### 2.3. Fabrication of Spray Dried (SD) L-Dopa Powders

SD L-dopa powders were prepared using the micro-fluidic spray dryer (MFSD), as described in the previous work [29]. Briefly, the feed solution (400 mL) was kept in the reservoir connected to the nozzle (orifice diameter = 75 μm), before being atomized into small droplets to the drying chamber, with inlet and outlet temperatures of 185 ± 3 °C and 88 ± 3 °C, respectively. The hot air flow rate was 280 ± 5 L/min. The obtained powders, with yields of approximately 85%, were stored in desiccator at 25 ± 2 °C for further characterization. The sample was named by SD-LxHy, SD-LxPy, or SD-LxCDy, where x:y was the mass ratio of L-dopa to excipient.

### 2.4. Particle Characterization

#### 2.4.1. Particle Size and Morphology

The particle morphology was characterized by scanning electron microscope (SEM, SU1510, Hitachi High Technologies Corporation, Tokyo, Japan). The particle size and distribution were acquired by analyzing SEM images containing over 500 particles using ImageJ. The equivalent particle geometric diameter (*D_g_*) was characterized as the minimum diameter of the circumscribed circle of projection surface of individual particles. The particle aerodynamic diameter (*D_a_*) was calculated by Equation (1):(1)$$D_{a}=D_{v}\sqrt{\frac{\rho }{\gamma \rho_{0}}}$$
where *D_V_* is the particle volumetric diameter, *ρ* is the particle density, *ρ*_0_ is the unit particle density, and *γ* is the particle dynamic shape factor. For quasi-spherical particles, it was assumed that *D_V_* ≈ *D_g_* and *γ* ≈ 1 [24].

#### 2.4.2. Moisture Content, Powder Density and Flowability

The moisture content of the as-prepared powders was measured gravimetrically. Approximately 0.1 g of sample was placed in an air oven at 105 °C for 3 h. The moisture content was calculated by Equation (2):(2)$$Moisture=\left[\frac{W_{0}-W_{1}}{W_{0}}\right]\times 100\%$$
where *W*_0_ and *W*_1_ represent the original and treated samples, respectively.

The bulk density (*ρ_b_*) was determined by loosely filling a weighed amount of powder (0.1–0.8 g) into a 5 mL measuring cylinder; the tapped density (*ρ_t_*) was the ultimate (equilibrium) density that would not change with the further increase of tap numbers [30]. The average density of samples was recorded based on three repeat measurements. The powder flowability was characterized by Carr’s index (*CI*), based on Equation (3):(3)$$CI=\left[\frac{\rho_{t}-\rho_{b}}{\rho_{t}}\right]\times 100\%$$
where *ρ_b_* and *ρ_t_* represent the bulk and tapped densities, respectively. Generally, *CI* < 25% indicates an acceptable particle flowability, while *CI* > 25% means poor flowability [31].

#### 2.4.3. Powder X-ray Diffraction (XRD)

Powder X-ray diffraction (XRD) characterization was performed with a desktop diffractometer (Cu-Kα, D2 PHASER, Bruker, Germany), which operated at 30 kV and 10 mA. The scanning range was 5°–90° (2θ basis), with a step size of 0.02° and 0.3 s for each step. Each experiment was repeated three times.

#### 2.4.4. Thermal Analysis

Differential scanning calorimetry (DSC) analyses were performed with a Mettler Toledo DSC 3+ instrument. Samples (~5 mg) were loaded onto an Al pan (40 μL, Cat. 00027331, Mettler Toledo) with a one-hole lid and analyzed in a temperature range of 5–350 °C at a heating rate of 10 °C/min under a nitrogen flow (50 mL/min).

#### 2.4.5. Fourier Transform Infrared (FT-IR)

Fourier transform infrared (FT-IR) spectra were obtained by using a Bruker tensor 27 spectrometer (Bruker, Germany) with a resolution of 4 cm^−1^ and wavenumber range of 400 to 4000 cm^−1^ using KBr pellets at room temperature.

### 2.5. In Vitro L-Dopa Release Study

In vitro L-dopa release study was performed using a Franz diffusion cell (12 mm, 15 mL volume, TP-6P, Tianjin Pharmacopoeia Standard Instrument Factory, China). The SNF was filled into receptor cell and stirred (200 rpm) at 34 °C. Accurately weighed 10 mg of L-dopa-doped powders was uniformly dispersed on the cellulose acetate membrane (0.45 μm, Xingya Purification Material Factory, Shanghai, China) inserted between the donor and receptor cells, followed by the addition of 0.5 mL SNF to the donor cell, in order to facilitate uniform spreading of the powder over the whole membrane surface and wetting [32]. At scheduled time intervals, 200 μL of samples were withdrawn from the receptor cell and replaced with the fresh SNF of same volume. Ultraviolet–visible spectrophotometer (DR6000, HACH, Washington, DC, USA) was used to quantitative analysis the L-dopa content at λ = 280 nm, and the calibration curve of L-dopa was linear over the concentration range of 1–50 μg∙mL^−1^ (R^2^ > 0.999). Each sample was measured three times.

### 2.6. In Vitro Mucoadhesion Test

Evaluation of mucoadhesive properties was carried out by measuring the particle–mucin adhesive force using an atomic force microscope (AFM, Dimension Icon, Bruker, Billerica, MA, USA), as previously reported [24]. Briefly, the Si_3_N_4_ probes (nominal spring constant = 0.58 nN/m, BudgetSensors, Innovative Solutions Bulgaria Ltd., Bulgaria) were covered by the simulated nasal mucus (95% water, 3% mucins, and 2% NaCl) [33]. The mucus-covered probe was then contacted with the microparticles for 10 s under 25% RH at 25 °C. Based on the force–distance curves, the particle–mucin adhesive force was calculated through the software of NanoScope Analysis 1.7 (Bruker, USA). A total of 300 randomly tests were performed for each sample.

### 2.7. In Vitro Nasal Delivery and Deposition Performance of L-Dopa Powders

A self-developed 3D-printed cast of nasal cavity, based on CT tomography of a male adult’s nose, was applied to evaluate the nasal delivery and deposition performance of L-dopa powders. The model is assembled virtually by five detachable compartments as nasal vestibule, lower turbinate, middle turbinate, olfactory region, and nasopharynx (Figure 5). The internal wall of each compartment was coated with 1% (*v*/*v*) Tween 20 in methanol, allowing for particle deposition [34,35]. During the deposition assessment, 20 mg of powder was administered into the cast using a simplified 3D-printed nasal powder delivery device (Appendix A) through a breathing simulator that enabled driving flow rates up to 50 L/min within 3 s [36,37]. The administration angle was set at 60° from horizontal, and the tip of the device was inserted at a distance of 5 mm. The delivery device was weighed before and after actuation to determine the actual emitted sample amounts. After administration, the nasal cavity model was disassembled, and all the compartments were rinsed with deionized water to dissolve deposited powders. The L-dopa content was determined spectrophotometrically at λ = 280 nm. Accordingly, the deposition profiles at different positions were acquired. All measurements were performed in triplicate.

### 2.8. Statistical Analysis

The model-independent similarity factor of *f*_2_ was calculated according to Equation (4) and employed to evaluate the drug release profiles of different samples.
(4)$$f_{2}=50\times \log\left\{{\left[1+\left(1/n\right)\overset{n}{\underset{t=1}{\sum }}{\left(R_{t}-T_{t}\right)}^{2}\right]}^{-0.5}\times 100\right\}$$
where *R_t_* and *T_t_* are the dissolution value at time point *t* of the reference and test product, respectively. The release profiles of different samples could be considered to be similar when *f*_2_ > 50 [28,38].

## 3. Results and Discussion

### 3.1. Particle Size, Morphology, Density, and Moisture Content

Uniform L-dopa/excipient powders were successfully prepared by spray drying the feed solution (Table 1) via MFSD. Unlike commercial L-dopa exhibiting irregular shapes, with polydisperse sizes ranging from several micrometers to hundreds of micrometers (Figure 1a), SD-L-dopa microparticles presented a spherical morphology with a smooth surface (Figure 1b) and extremely narrow size distribution (25.61 ± 0.46 μm) (Table 2). The addition of HPMC, PVP, or CD generated wrinkled microparticles with different sizes and surface textural structures, depending on the excipient type (Figure 1d–f). HPMC tended to produce microparticles with the highest degree of surface depression than the PVP- and CD-involved microparticles, whereas the CD-involved microparticles were endowed with the best sphericity. During drying process, the viscoelastic shell of the droplet formed by the precipitation of the precursor ingredients was subjected to buckling under capillary force, induced by the continuous drying, thus leading to wrinkled microparticles [26,39]. The SD microparticles containing excipient also showed an extremely narrow size distribution (Table 2). It was noted that the drug/excipient mass ratio had a very limited influence on the particle geometric size when using the same kind of excipient, e.g., 43.39 ± 1.41, 44.40 ± 1.43, and 45.70 ± 1.68 μm for SD-L2H1, -L1H1, and -L1H2, respectively, whereas the effect of the excipient type on the particle geometric size was more apparent and changed, in the order of HPMC > PVP > CD, at the same excipient content (Table 2), e.g., 40.81 ± 1.43 μm for SD-L1P2 and 38.91 ± 1.07 μm for SD-L1CD2, possibly due to the different time for shell formation of droplet containing various types of excipient [40]. The polymeric molecule of HPMC and PVP tended to precipitate within a shorter time than that of CD, resulted from their relatively large hydrodynamic size and low solubility [40,41,42], thus accelerating the formation of the shell and, consequently, the larger SD microparticle.

The particle bulk density and tapped density gradually decreased as the excipient content increased for each kind of excipient (Table 2), mainly due to the fact that the density of L-dopa (~1.5 g/cm^3^) is greater than that of HPMC (~1.39 g/cm^3^), PVP (~1.144 g/cm^3^), and CD (~1.4 g/cm^3^). Compared to HPMC- (CI = 29.34 ± 2.70% ~ 37.32 ± 2.51%) and PVP-involved (CI = 26.31 ± 2.10% ~ 34.51 ± 2.74%) microparticles, the lower CI value (varied between 12.10 ± 1.86% and 17.91 ± 1.06%) of CD-involved microparticles revealed its better flowability (Table 2). Moreover, SD particles containing HPMC and PVP exhibited higher moisture contents (i.e., between 5.26 ± 0.17 wt% and 6.33 ± 0.20 wt%) than CD-involved microparticles of 4.33 ± 0.28 wt% and 4.57 ± 0.43 wt%, e.g., 5.38 ± 0.65 wt%, 6.33 ± 0.20 wt% and 4.33 ± 0.28 wt% for SD-L1H2, -L1P2 and -L1CD2, respectively. During drying, the HPMC and PVP of higher molecular weights migrated more slowly toward the inside of droplet than CD, and they tended to precipitate to form a shell on the droplet surface. The formed shell jeopardized water evaporation and led to a higher moisture content [40].

### 3.2. Particle Crystal Property and Storage Stability

The particle’s crystal properties were characterized by XRD (Figure 2A) and DSC (Figure 2B). The as-supplied L-dopa showed sharp peaks between 18.89° and 29.12°, due to its crystalline nature (Figure 2A (a)). In addition, the as-supplied L-dopa showed only one endothermic peak at 297 °C, corresponding to its melting point (Figure 2B (a)). Compared to the as-supplied L-dopa, the sharp crystalline peaks disappeared in the XRD pattern of SD-L-dopa (Figure 2A (e)), indicating its amorphous state. Meanwhile, the recrystallization peak at 180.83 °C and melting peak at 291 °C in the DSC curve (Figure 2B (e)) also confirmed its amorphous state [43]. This amorphous state of L-dopa could have resulted from the extremely fast evaporation rate of the droplet during spray drying, which allowed insufficient time for L-dopa crystal formation [44,45]. All three raw excipients were amorphous (Figure 2A (b–d)), due to their high glass transition temperature (Tg), i.e., 155.86 °C for HPMC (Figure 2B (b)), 160.0 °C for PVP (Figure 2B (c)), and 213 °C for CD (Figure 2B (d)) [46]. After co-spray drying L-dopa with excipients, no XRD (Figure 2A (f–n)) or apparent exothermic (Figure 2B (f–n)) peaks were observed, indicating the formation of L-dopa-doped amorphous solid dispersions [44]. Compared to the SD-L-dopa, the shift of recrystallization and melting peaks in co-spray dried sample (Figure 2B (f–n)) confirmed the intermolecular interactions between the L-dopa and excipients, which is further discussed in the FT-IR section. Such a considerable effect of these excipients on the anti-crystallization of L-dopa could be explained with two aspects. On the one hand, according to the Williams–Landel–Ferry theory, the crystallization rate is a function of the difference between the Tg of the materials and process temperature (Tp) (Tp-Tg) [47]. Excipients of high Tg intended to accelerate the drying rate of the droplet, thereby restricting the time for L-dopa nucleation. On the other hand, the cross-linked chains of HPMC and PVP, as well as the interior cavity of CD, could act as physical barriers to reduce the molecular mobility of L-dopa nucleation, thus inhibiting its crystallization.

It was reported that amorphous substances possessed high energy and were inclined to transform to a more stable crystalline state during storage, which would likely compromise the re-dissolubility and bioavailability of the drugs [48]. After storage at 22 °C under 18% RH in a Ziplock bag for 24 h, few tiny crystals appeared on the surface of SD-L-dopa (Figure 1c); accordingly, the diffraction peaks appeared in the XRD spectrum (Appendix A), indicating the crystallization of amorphous L-dopa (Figure 1b). By contrast, there were no changes in the particle morphology (Appendix A) and crystal properties (Appendix A) of the SD samples, with a relatively small proportion (1/3) of the excipients after storage at 22 °C under 18% RH for one month. This suggests that the excipients continuously play a critical role in preventing the crystallization of L-dopa during storage. Apart from the possible reasons mentioned above, such an excellent anti-crystallization effect might be due to the specific interactions of the excipients with L-dopa, upon which the chemical potential necessary for nucleation of L-dopa was highly reduced [44].

To verify the possible interactions between the L-dopa and excipients, FT-IR characterization of typical samples was carried out. As shown in Figure 2C (a), raw L-dopa showed feature peaks centered at 3211.10 cm^−1^ assigned to the stretching vibrations of -NH_2_ groups and 1655.80 cm^−1^ for stretching vibrations of C=O. Pure HPMC and CD showed prominent absorption bands at 3471.18 (Figure 2C (b)) and 3422.75 cm^−1^ (Figure 2C (d)), respectively, which are mainly attributed to the stretching vibrations of the -OH groups. For pure PVP (Figure 2C (c)), peaks at 1662.81 cm^−1^, attributed to the stretching vibrations of C=O, were observed; meanwhile, the broad band at 3453.07 cm^−1^, corresponding to -OH groups, indicated the presence of water, which is in good agreement with the results of DSC (Figure 2B (c)). Compared to raw L-dopa and HPMC, for SD-L2H1 (Figure 2C (f)), SD-L1H1 (Figure 2C (g)), and SD-L1H2 (Figure 2C (h)), the stretching vibration peak of the -OH group in HPMC downshifted from 3477.18 to 3451.30 (Δν = 25.88 cm^−1^), 3444.72 (Δν = 32.46 cm^−1^), and 3472.17 cm^−1^ (Δν = 5.01 cm^−1^), respectively. Meanwhile the C=O of L-dopa migrated to 1639.22 (Δν = 16.58 cm^−1^), 1649.98 (Δν = 5.82 cm^−1^), and 1640.60 cm^−1^ (Δν = 15.2 cm^−1^), respectively, which evidenced the hydrogen bonds formed between L-dopa and HPMC after spray drying [49]. For SD-L2P1 (Figure 2C (i)), SD-L1P1 (Figure 2C (j)), and SD-L1P2 (Figure 2C (k)), the stretching vibration peak of the C=O groups in PVP downshifted from 1662.81 to 1648.28 (Δν = 14.53 cm^−1^), 1652.75 (Δν = 10.06 cm^−1^), and 1657.15 cm^−1^ (Δν = 5.66 cm^−1^), respectively, which also indicated the hydrogen bonds formed between PVP and L-dopa through interactions with the -NH_2_ group of L-dopa [50]. With regard to SD-L2CD1 (Figure 2C (l)), SD-L1CD1 (Figure 2C (m)), and SD-L1CD2 (Figure 2C (n)), the -OH group migrated from 3422.75 to 3414.31 (Δν = 8.44 cm^−1^), 3404.01 (Δν = 18.74 cm^−1^), and 3407.24 cm^−1^ (Δν = 15.51 cm^−1^), respectively; additionally, the C=O of L-dopa red-shifted, which also demonstrate the hydrogen bonds between L-dopa and CD. Such obvious hydrogen bonds formed between excipients and L-dopa could be another reason for the crystallization inhibition.

### 3.3. In Vitro L-Dopa Release Behavior

The in vitro release profiles of L-dopa from the SD microparticles containing the excipients were assessed using a vertical Franz diffusion cell. Although SD-L-dopa was of the amorphous state, its dissolving rate was apparently lower than that of raw L-dopa (Figure 3). This could be because SD-L-dopa was hygroscopic and tended to be agglomerated into big lump, which impeded the wetting process of the sample, thus decelerating its dissolution. By comparison with the raw L-dopa and SD-L-dopa, the excipient-involved SD microparticles were generally of an increased release rate regarding L-dopa, however, to different extent (Figure 3), which correlated well with the wetting process facilitated by the hydrophilic excipient. Once dispersed in the release medium, a hydration layer formed, covering the microparticle surface, due to the good wettability of the excipient; meanwhile, the amorphous drug molecules dissolved and released to the medium [28,51]. Using different amounts of HPMC as an excipient, the L-dopa release rates were quite similar, as indicated by 58.92 < *f*_2_ < 70.39, which was calculated according to the Equation (4), suggesting the limited influence of HPMC content on the L-dopa release rate. Increasing the PVP content induced a slightly faster L-dopa release, as revealed by the *f*_2_ = 47.75 of SD-L2P1 and -L1P1; however, further addition of PVP did not significantly accelerate the L-dopa release rate, with *f*_2_ = 64.24 of SD-L1P1 and -L1P2. As for the CD-involved powders, a gradually elevated release rate of L-dopa was observed for powders of higher CD content (23.75 < *f*_2_ < 42.48), which could be ascribed to the fact that more CD content accelerated the erosion of powder, thereby increasing the release rate of L-dopa [52].

With regard to the effect of the excipient type on the L-dopa release, it was found that there was no significant difference in the release rate of L-dopa for the SD samples of the drug/excipient ratios 2:1 and 1:1 (Figure 3A,B), as indicated by 51.51 < *f*_2_ < 73.49. It implies that the influence of the excipient type on the L-dopa release was limited when the excipient content was relatively low. By contrast, with the drug/excipient ratios of 1:2, L-dopa was released more rapidly from the CD-involved microparticles than HPMC and PVP. Particularly, more than ~83% of the total L-dopa dose was released in 60 min for SD-L1CD2, much higher than SD-L1P2 (~71%) and SD-L1H2 (~64%) (Figure 3C). This could be ascribed to the higher solubility of CD, as compared to that of HPMC and PVP, i.e., ~65, 5, and 5 g/100 mL H_2_O, respectively.

### 3.4. In Vitro Mucoadhesive Property

The in vitro mucoadhesion test was performed using AFM by fully covering the AFM probe with the mucus, in order to adhere to the surface of SD microparticles. Although this method is still arguable, in terms of the correlation between in vitro and in vivo, its feasibility for comparing the mucoadhesive property of various samples has been validated [53,54,55]. Figure 4 displays the representative adhesion force–distance curves of different SD powders. SD-L-dopa powder showed an adhesion force similar to that of the raw L-dopa (23.16 ± 2.69 nN), thus indicating the neglectable influence from the crystalline state of L-dopa. HPMC-involved microparticles showed adhesion forces of 63.54 ± 7.19 nN, 117.43 ± 12.32 nN, and 269.23 ± 7.91 nN for SD-L2H1, -L1H1, and -L1H2, respectively. By using PVP as the excipient, the adhesion forces were 26.94 ± 9.28 nN, 69.81 ± 8.20 nN, and 110.66 ± 13.36 nN for SD-L2P1, -L1P1, and -L1P2, respectively. Obviously, the adhesion force increased with more HPMC or PVP content, probably due to the fact that hydrated polymer chains of HPMC and PVP could interpenetrate with mucus glycoproteins [56], thereby enhancing the interactions between the microparticles and nasal mucosa via van der Waals forces, hydrogen bonds, hydrophobic forces, or/and electrostatic forces [56,57]. In particular, HPMC had a more considerable effect on the adhesion force than PVP with the same drug/excipient ratio, mainly due to the longer polymeric chains and more hydrogen bonds donor/acceptor of HPMC, which could facilitate its interaction with mucin to form a strengthened network. CD had a very limited influence on the adhesion force of the SD microparticles, as evidenced by the value of ~24.34 nN, regardless of the mass ratio of L-dopa to CD, possibly due to the poor interaction of CD and mucus. These results demonstrated that the SD microparticles’ mucoadhesion could be tuned by rationally selecting different types and contents of excipient for achieving a prolonged residence time in the nasal cavity.

### 3.5. In Vitro Nasal Deposition Performance

Although there has been no comprehensive international consensus regarding the relationship between aerosol characteristics and deposition sites of the nasal powders, the deposition performance should be carefully evaluated, in conjunction with the particle formulation, structural attributes, and administration parameters, etc., in the early phase of the product development of nasal powders, in order to achieve the expected therapeutic outcome [58]. To evaluate the in vitro deposition performance of nasal powders, a 3D-printed nasal cast has been created, according to the CT scans of an adult human nose (Figure 5A). The nasal valve (Figure 5B), with a minimum coronal cross-sectional area (Area_min_) of approximately 20 mm^2^, is located beyond the nasal vestibule and at the anterior aspect of the inferior turbinate, which is a natural obstruction for inhaled particles, resulting from the narrowing and twisting of the airway in this region. The nasal cast (Figure 5C) was divided into five representative parts, i.e., the nasopharynx, nasal vestibule, olfactory region, and middle and inferior turbinate (Figure 5D (1–5)). Among them, the olfactory region has the potential for direct nose-to-brain drug delivery, while the turbinate represents the targeted area to achieve local drug effects or systemic drug bioavailability [59].

SD-L1H2, SD-L1P2, and SD-L1CD2, with relatively rapid drug release and large adhesion forces, were selected for the in vitro nasal deposition tests. Figure 6 shows the deposition fractions in different regions of the nasal cast. Overall, all the samples exhibited approximately 99% doses delivered to the nasal cavity under a mimic breath flow of 50 L/min. The total dose recovery ratios calculated were 92.80 ± 6.01% for SD-L1H2, 88.24 ± 2.04% for SD-L1P2, and 90.56 ± 5.43% for SD-L1CD2 (Appendix A). More importantly, these samples showed good deposition fractions, both for the turbinate and olfactory regions, i.e., approximately 30% and 14%, respectively (Figure 6), rendering the high potential for effective L-dopa delivery.

Under the action of airflow, the powder movement, impaction, and adhesion behaviors were the primary determinants of the deposition regions in the nasal cavity. It is noted that the vastly different adhesion force of the tested samples had almost no influence on the deposition fraction. As for the particle impaction that could occur whenever the air stream carrying the particles changes its direction along the intranasal passage, the impaction probability is proportional to $Ud^{2}\sin\theta /R$, where *θ* is the angle of the airway bend (Appendix A), *U* is the airstream velocity, *d* is the particle aerodynamic diameter, and *R* is the airway radius [60]. Since *θ* and *U* were kept constant in this study, d was the determinant factor of the particle impaction. As the SD-L1H2, -L1P2, and -L1CD2 had similarly calculated aerodynamic diameters (D50), according to Equation (1), i.e., 27.04 ± 1.00, 31.08 ± 1.09, and 28.06 ± 0.77 μm, respectively (Table 2), they likely moved in an analogous trajectory and tended to impact similar areas of the nasal cavity. Based on their quite similar deposition fractions, it was reasonably inferred that even the lowest adhesion force (~26 nN) of the SD-L1CD2 could be sufficient for its deposition after impaction. More systematical investigation on the effects of particle aerodynamic diameter and mucosal adhesion force on the deposition performance of nasal powders is being conducted and will be reported in future.

## 4. Conclusions

A series of L-dopa-doped powders, potentially for nasal delivery, were successfully fabricated by co-spraying L-dopa with the excipients of HPMC, PVP, or CD via MFSD. The excipient-involved powders were of particle sizes ranging from 36.92 to 45.70 μm and showed extremely narrow size distribution. Compared with HPMC- and PVP-involved powders, the CD-involved powders presented excellent flowability, as indicated by the CI between 12.10% and 17.91%, mainly due to their size uniformity and much better sphericity. Meanwhile, the doped L-dopa was amorphous, due to the high Tg of those excipients acting as physical barriers, which not only reduced the molecular mobility of L-dopa, but also restricted its nucleation. The addition of excipient, regardless of its type, could accelerate the L-dopa release to a different extent, which correlated well with the wetting process facilitated by the hydrophilic excipient. SD-L1CD2 showed the rapidest release rate, i.e., ~83% of L-dopa released in 60 min, compared to 37% for raw L-dopa. Moreover, HPMC could more apparently improve the particle mucoadhesion than PVP and CD, with the respective adhesive forces of ~269, 111, and 26 nN for SD-L1H2, -L1P2, and -L1CD2, mainly due to the relatively higher viscosity and cross-linked polymeric structure of HPMC. Although these three samples had remarkable differences in adhesion force, the deposition fractions in olfactory region were almost the same (~14%), probably ascribable to their quite similar particle aerodynamic diameter (~30 μm), which played a crucial role in the deposition efficiency. The formulation design and in vitro evaluation methods of L-dopa-doped powder presented in this study provide guidance for the development of nasal powder products.

## Figures and Tables

**Figure 1 pharmaceutics-14-01384-f001:**
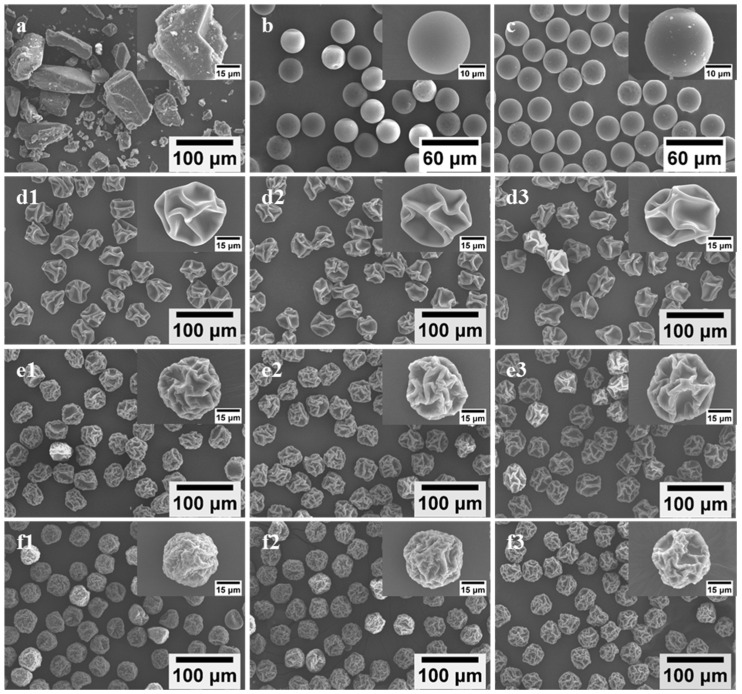
SEM images of raw L-dopa (**a**), SD-L-dopa (**b**), and SD-L-dopa after storage (**c**); SD-L2H1 (**d1**), SD-L1H1 (**d2**), and SD-L1H2 (**d3**); SD-L2P1 (**e1**), SD-L1P1 (**e2**), and SD-L1P2 (**e3**); SD-L2CD1 (**f1**), SD-L1CD1 (**f2**), and SD-L1CD2 (**f3**).

**Figure 2 pharmaceutics-14-01384-f002:**
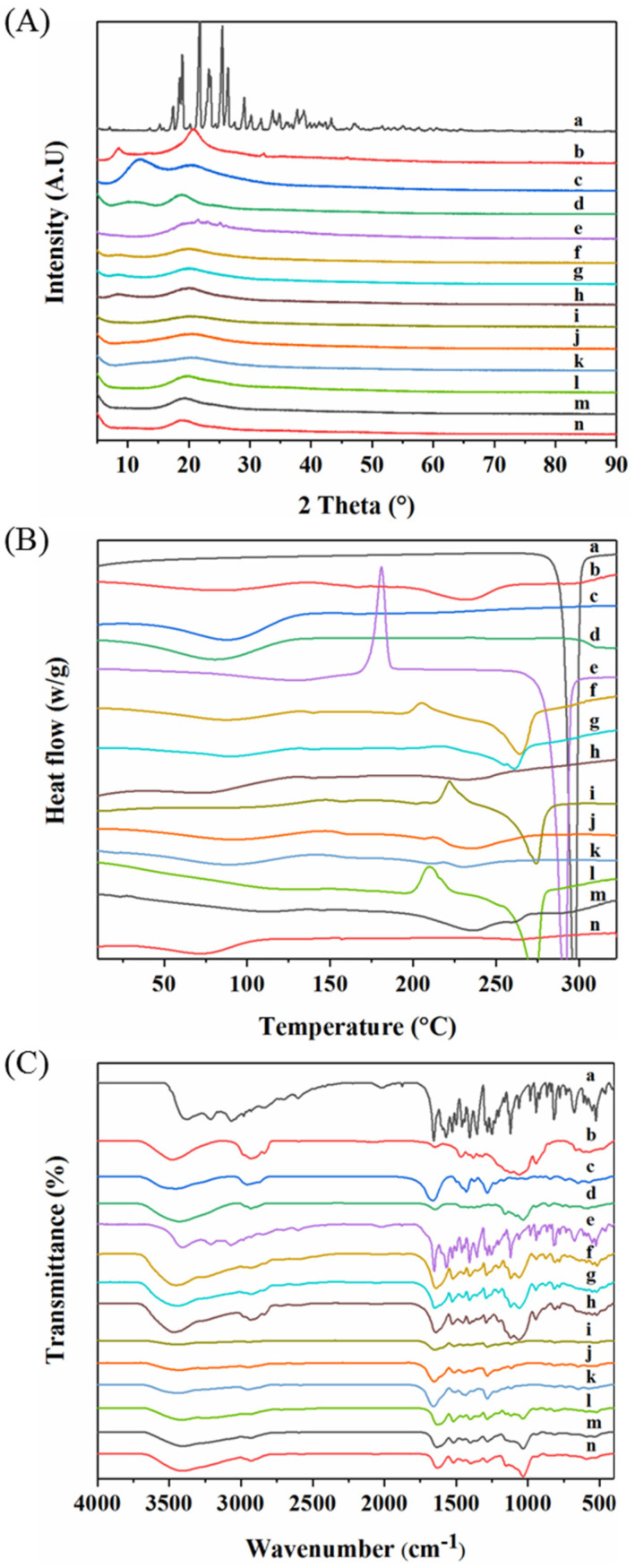
XRD patterns (**A**), DSC curves (**B**), and FT-IR spectra (**C**) of raw L−dopa (a), raw HPMC (b), raw PVP (c), raw CD (d), SD−L−dopa (e), SD−L2H1 (f), SD−L1H1 (g), SD−L1H2 (h), SD-L2P1 (i), SD−L1P1 (j), SD−L1P2 (k), SD−L2CD1 (l), SD−L1CD1 (m), and SD−L1CD2 (n), respectively.

**Figure 3 pharmaceutics-14-01384-f003:**
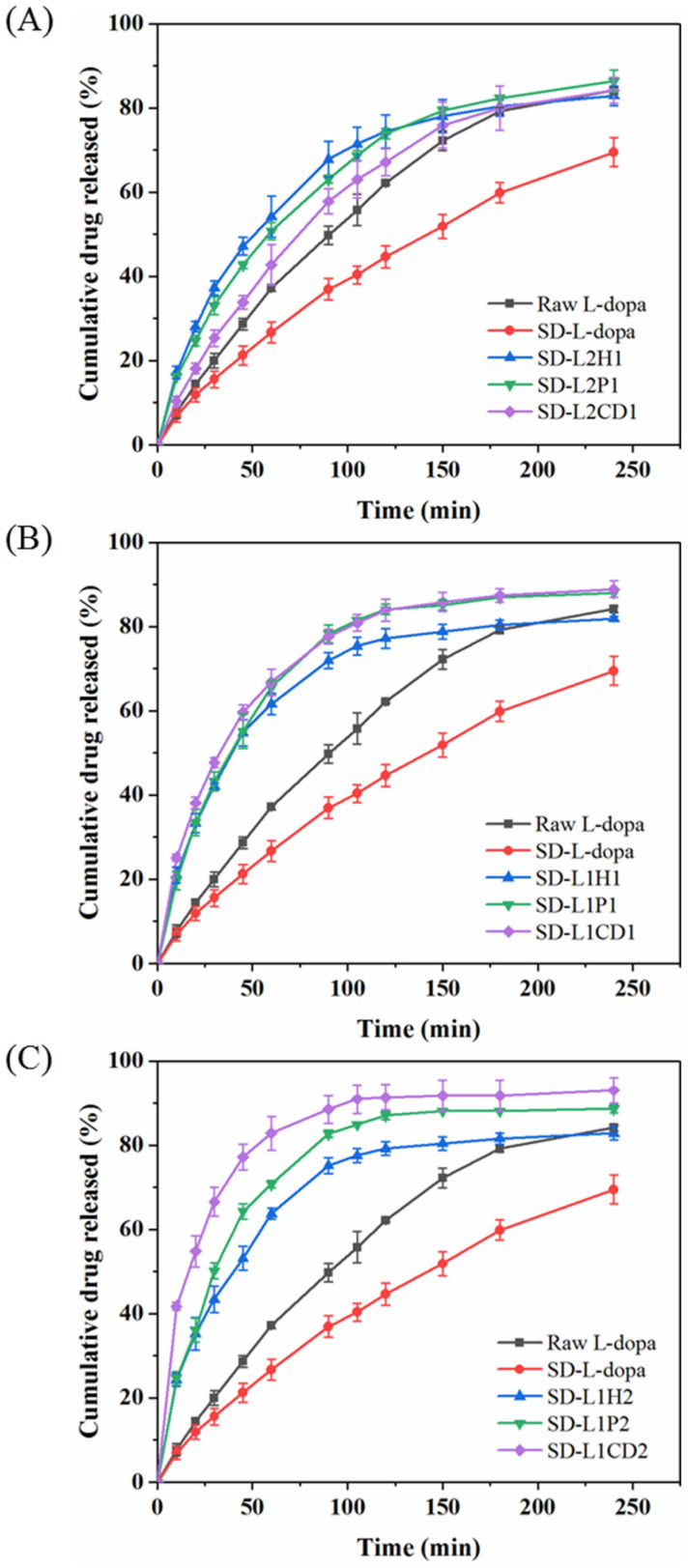
In vitro release profiles of L-dopa from raw L-dopa, pure SD-L-dopa, and different SD L-dopa-doped powders, with drug/excipient mass ratios of 2:1 (**A**), 1:1 (**B**), and 1:2 (**C**), respectively.

**Figure 4 pharmaceutics-14-01384-f004:**
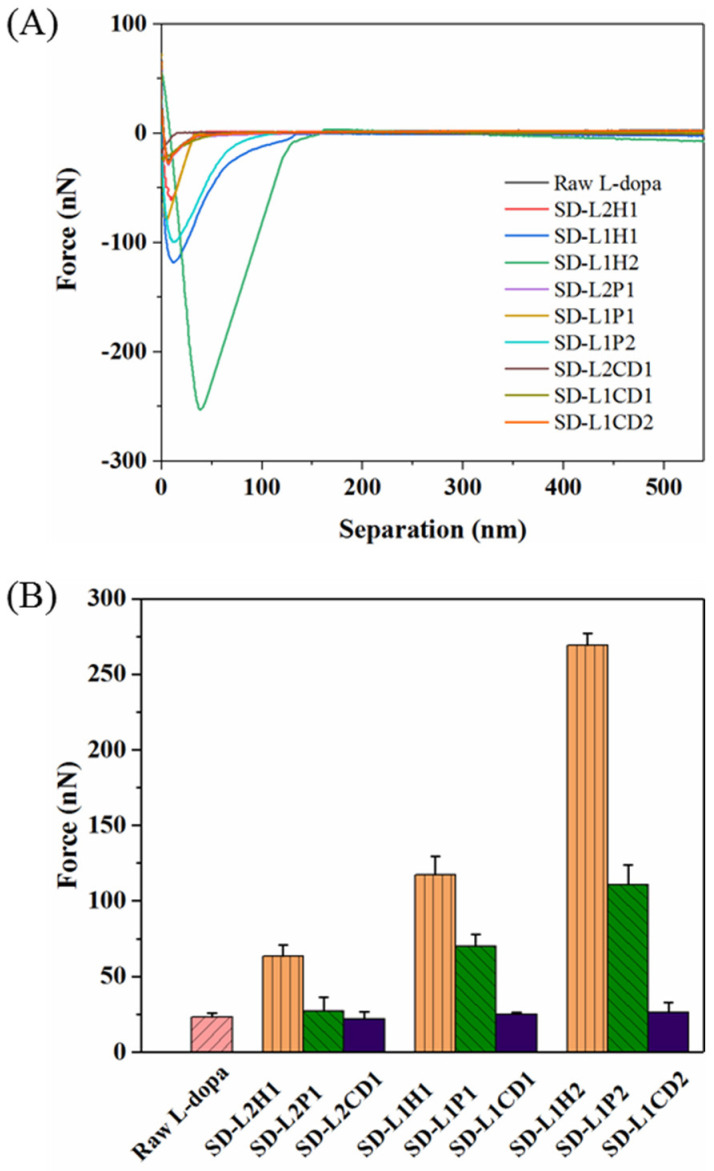
Force–distance curves (**A**) of raw L-dopa and different SD L-dopa-doped powders, as characterized using AFM, as well as the maximum values of their adhesion force (**B**).

**Figure 5 pharmaceutics-14-01384-f005:**
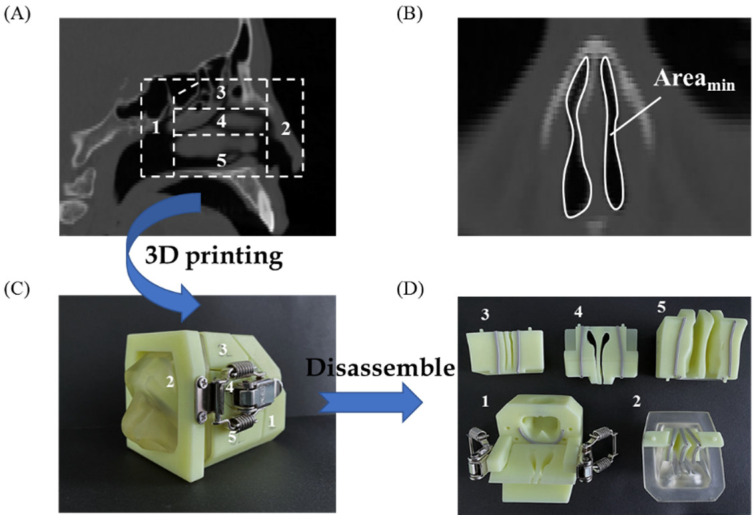
CT-scan image of an adult human nose (**A**); the nasal valve with the minimum coronal cross-section area (Area_min_ ~ 20 mm^2^) (**B**); a 3D-printed nasal cast based on the CT-scan (**C**); the disassembled part of the nasal cast (**D**): 1, nasopharynx; 2, nasal vestibule; 3, olfactory region; 4, middle turbinate; 5, inferior turbinate.

**Figure 6 pharmaceutics-14-01384-f006:**
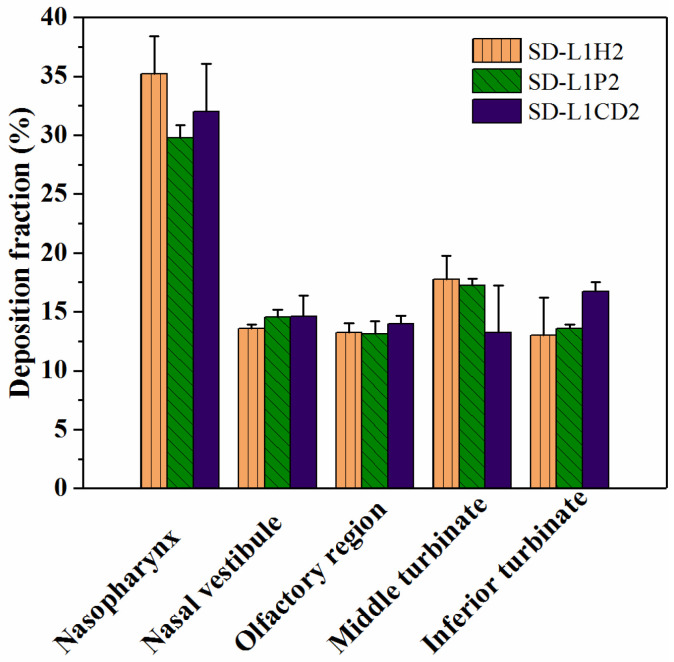
Regional deposition fractions of SD-L1H2, SD-L1P2, and SD-L1CD2.

**Table 1 pharmaceutics-14-01384-t001:** Formulation of feed solutions for spray drying.

Sample	L-Dopa (g/100 g Solution)	HPMC (g/100 g Solution)	PVP (g/100 g Solution)	CD (g/100 g Solution)	Total Solid Content (wt%)
L-dopa	0.3	/	/	/	0.3
L2H1	0.4	0.2	/	/	0.6
L1H1	0.3	0.3	/	/	0.6
L1H2	0.2	0.4	/	/	0.6
L2P1	0.4	/	0.2	/	0.6
L1P1	0.3	/	0.3	/	0.6
L1P2	0.2	/	0.4	/	0.6
L2CD1	0.4	/	/	0.2	0.6
L1CD1	0.3	/	/	0.3	0.6
L1CD2	0.2	/	/	0.4	0.6

**Table 2 pharmaceutics-14-01384-t002:** Summary of the physical properties of the various SD powders.

Sample	Particle Geometric Diameter (*D_g_*, μm)	Moisture Content (wt%)	Bulk Density (*ρ_b_*, g/cm^3^)	Tapped Density (*ρ_t_*, g/cm^3^)	Carr’s Index (CI, %)	Calculated Particle Aerodynamic Diameter (*D_a_*, μm)
SD-L-dopa	25.61 ± 0.46	/	/	/	/	/
SD-L2H1	43.39 ± 1.41	5.68 ± 0.25	0.34 ± 0.01	0.49 ± 0.01	30.03 ± 1.94	30.38 ± 0.99
SD-L1H1	44.40 ± 1.43	5.73 ± 2.07	0.29 ± 0.01	0.47 ± 0.01	37.32 ± 2.51	30.44 ± 0.98
SD-L1H2	45.70 ± 1.68	5.38 ± 0.65	0.25 ± 0.02	0.35 ± 0.02	29.34 ± 2.70	27.04 ± 1.00
SD-L2P1	40.27 ± 1.06	5.26 ± 0.17	0.50 ± 0.01	0.75 ± 0.02	32.61 ± 1.00	34.88 ± 0.92
SD-L1P1	40.99 ± 1.35	6.01 ± 0.63	0.45 ± 0.02	0.68 ± 0.02	34.51 ± 2.74	33.81 ± 1.11
SD-L1P2	40.81 ± 1.43	6.33 ± 0.20	0.43 ± 0.02	0.58 ± 0.02	26.31 ± 2.10	31.08 ± 1.09
SD-L2CD1	36.92 ± 0.88	4.57 ± 0.43	0.57 ± 0.02	0.65 ± 0.02	12.10 ± 1.86	29.77 ± 0.71
SD-L1CD1	38.83 ± 0.95	4.33 ± 0.32	0.47 ± 0.01	0.54 ± 0.02	12.68 ± 2.52	28.54 ± 0.70
SD-L1CD2	38.91 ± 1.07	4.33 ± 0.28	0.42 ± 0.02	0.52 ± 0.02	17.91 ± 1.06	28.06 ± 0.77

## Data Availability

Not applicable.

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
