# Peer review of "Spray Dried Levodopa-Doped Powder Potentially for Intranasal Delivery"

_pharmaceutics, 2022, doi:10.3390/pharmaceutics14071384_

Round 1
Reviewer 1 Report
The authors developed, by spray drying, a series of microparticles composed of L-dopa and excipients aiming to achieve controllable drug release and high nasal deposition efficiency. The study was very well executed, with relevant results corroborating with the conclusions. I think this work brings enough innovation and is relevant to this Journal. It could be recommended for publication in Pharmaceutics if the following concerns can be sufficiently addressed.
- What is the explanation for the obtained SD microparticles to have prolonged residence time, improved distribution and absorption of L-dopa in the nasal cavity? What are the critical attributes for nasal powder formulations, regarding particle size requirements?
- About the device that was used for delivery the nasal powder formulation: Where/How is the powder reservoir? Does it exist? or Does the device use capsules? Please add information about this subject.
- Line 77: “... Hydroxypropyl methyl cellulose (HPMC), polyvinylpyrrolidone (PVP) and CD were selected as excipients”. Please justify excipients selection, including its formulation function and suitability for nasal route.
- Itens 2.4.3, 2.4.4 and 2.4.5 need to be spaced.
- The difference between "equivalent particle geometric diameter (Dg)" and "particle aerodynamic diameter (Da)" was not clear. According to the values in Table 2, it can be seen that they do not have a linear correlation.
- About the tapped density: please discuss about it, since it is the indication of powder packing and interparticulate contact.
- Please add discussion about aerodynamic diameter and its suitability for nasal powder formulation.
- DSC curve of “as-supplied L-dopa” was included but not interpreted nor discussed. Please add the results and discussion in the manuscript text.
- Line 230: “Compared to HPMC- and PVP-involved microparticles, CD-involved microparticles showed better flowability”. Please include the Carr´s index value and its reference. What do you mean by “better flowability”? It´s vague. Define it and discuss accordingly.
- Line 243: “Pure SD-L-dopa did not exhibit significant crystalline peaks (Figure 2A-e) but exothermic peaks at 180.83 °C (Figure 2B-e), indicating its amorphous state”. Please re-write this sentence because DRX and DSC information together is unclear. In addition, DSC SD-L-dopa also exhibits an endothermic peak in the figure that is ignored in the text. Please include and discuss accordingly.
- Line 249: “After co-spray drying L-dopa with excipients, no XRD peaks (Figure 2A f-n) and apparent exothermic peaks (Figure 2B f-n) were observed, which implied the formation of L-dopa-doped amorphous solid dispersion”. DSC curves f, i and l (Figure 2B) show a low intensity exothermic peak between 200 – 250oC and also an endothermic peak. Please discuss these thermal events. Is there temperature shift for the thermal events? If yes, is there any implication for drug-excipient association?
- Line 262: “After storage at 22 °C under 18% RH for 24 h”. Please justify these conditions and the primary container of the spray-dryed powders.
- Line 267: “relatively small proportion (1/3) of the excipients after storage at 22 °C under 18% RH for…” Is the proportion (1/3) correct? This proportion was mentioned only in this line.
- Why was the release assay performed for 250 min? Is this the actual residence time of particles in the nasal cavity? It is interesting to add a reference.
- The authors comment that “SD particles containing HPMC and PVP exhibited higher moisture contents than CD-involved microparticles” and cited that HPMC and PVP are more hydrophilic. However, in the release assay, “with the drug/excipient ratios of 1:2, L-dopa was released more rapidly from the CD-involved microparticles than HPMC and PVP. This could be ascribed to the higher solubility of CD than that of HPMC and PVP, i.e., ~ 65, 5 and 5 g/100 mL H2O, respectively”. Could the authors better explain the difference between the hydrophilicity and solubility profiles?
- Regarding the "3D-printed quasi-real human nasal cast", do the authors think it would be possible to add mucus to better simulate real conditions? Could this change the deposition fractions in olfactory region profile?
- Suplementary material: Figure S4. Add conditions (temperature, humidity and any other information that could be necessary).
Reviewer 2 Report
This is an important study showing the potential of developing levodopa powder for nasal delivery. The manuscript is well written. The experiments were properly planned and conducted. The data were presented nicely with proper discussion. After addressing the following issues, this manuscript can be accepted in this journal if other requirements mentioned in the guidelines full-filled.
Specific comments
Line 95, please write city name to keep consistency with all other manufacturer’s name
Line 101, please correct ‘formula’ as ‘formulation’ in the sentence.
Line 108, why pure L-dopa feed solid concentration was 0.3% w/w different than other formulations?
Line 116, what was the RH of the desiccator?
Equation 3, please clarify everything in this equation as other equations.
Line 147, why zirconia pan was used in DSC while widely Al pan is used?
Line 185, correct though as through
Reviewer 3 Report
In this paper, the authors prepared through spray drying microparticles of levodopa using different carriers. The topic is interesting, the experimentation well conducted, the results well interpretated, and the methodology clearly described; therefore, the paper can be published following some revisions.
Some sentences have to be completed. See, for example, lines 47-49.
In the introduction, the authors listed some techniques used to prepared microparticles. Innovative supercritical fluids based techniques allow the attainment of powders with controlled dimensions and distributions. They should be named in the introduction (see, for example, doi: 10.1021/acs.iecr.5b03504 and doi: 10.1016/j.supflu.2014.07.001).
Some sentences are not clear. See, for example, lines 245-247 or line 321.
Regarding Figure 3, it is not clear to me why the cumulative drug release reached a value lower than 100%.
